# A Novel Cysteine Protease Inhibitor of *Naegleria fowleri* That Is Specifically Expressed during Encystation and at Mature Cysts

**DOI:** 10.3390/pathogens10040388

**Published:** 2021-03-24

**Authors:** Hương Giang Lê, A-Jeong Ham, Jung-Mi Kang, Tuấn Cường Võ, Haung Naw, Hae-Jin Sohn, Ho-Joon Shin, Byoung-Kuk Na

**Affiliations:** 1Department of Parasitology and Tropical Medicine, and Institute of Health Sciences, Gyeongsang National University College of Medicine, Jinju 52727, Korea; gianglee291994@gmail.com (H.G.L.); gjm9951001@hanmail.net (J.-M.K.); vtcuong241@gmail.com (T.C.V.); haungnaw23@gmail.com (H.N.); 2Department of Convergence Medical Science, Gyeongsang National University, Jinju 52727, Korea; 3Department of Microbiology, Ajou University College of Medicine, Suwon 16499, Korea; dkwjd0521@naver.com (A.-J.H.); hj35good@ajou.ac.kr (H.-J.S.); hjshin@ajou.ac.kr (H.-J.S.)

**Keywords:** *Naegleria fowleri*, cysteine protease inhibitor, encystation, cyst

## Abstract

*Naegleria fowleri* is a free-living amoeba that is ubiquitous in diverse natural environments. It causes a fatal brain infection in humans known as primary amoebic meningoencephalitis. Despite the medical importance of the parasitic disease, there is a great lack of knowledge about the biology and pathogenicity of *N. fowleri*. In this study, we identified and characterized a novel cysteine protease inhibitor of *N. fowleri* (NfCPI). NfCPI is a typical cysteine protease inhibitor belonging to the cystatin family with a Gln-Val-Val-Ala-Gly (QVVAG) motif, a characteristic motif conserved in the cystatin family of proteins. Bacterially expressed recombinant NfCPI has a dimeric structure and exhibits inhibitory activity against several cysteine proteases including cathespin Bs of *N. fowleri* at a broad range of pH values. Expression profiles of *nfcpi* revealed that the gene was highly expressed during encystation and cyst of the amoeba. Western blot and immunofluorescence assays also support its high level of expression in cysts. These findings collectively suggest that NfCPI may play a critical role in encystation or cyst formation of *N. fowleri* by regulating cysteine proteases that may mediate encystation or mature cyst formation of the amoeba. More comprehensive studies to investigate the roles of NfCPI in encystation and its target proteases are necessary to elucidate the regulatory mechanism and the biological significance of NfCPI.

## 1. Introduction

*Naegleria fowleri* is a free-living amoeba that is ubiquitously distributed in diverse natural environments including freshwaters and soil [1]. However, it can infect humans and cause a rare but fatal brain infection known as primary amoebic meningoencephalitis (PAM) [2]. Once the parasite enters the human body via the nasal route, it crosses the olfactory neuroepithelia, travels to the central nervous system, and eventually reaches the brain. The clinical manifestations of *N. fowleri* infection normally appear within 2–8 days after infection, which include symptoms of fever, severe headache, nausea, chills, neck stiffness, seizures, and coma and eventually result in fatal PAM characterized by necrotizing, fulminant, and hemorrhagic meningoencephalitis [2,3]. The progression of PAM is rapid, and usually results in death within 14 days. Until 2018, 145 cases were reported in the United States, but the mortality rate was extremely high, at 97% [4]. Concern for expansion of PAM associated with climate changes has also increased in recent years [5]. Cases of PAM induced by *N. fowleri* infection have also been reported in diverse global areas, including Asia, Africa, Europe, and South America [6].

Pathogenesis of PAM has not been fully understood, but it has been proposed that *N. fowleri* may induce cell death and an inflammatory response in host brain via two different mechanisms, contact-dependent and contact-independent mechanisms [3,7,8]. The contact-dependent mechanism leads to direct host cell damage via active trogocytosis by *N. fowleri* trophozoites, in which food-cup structure (amoebastome) is involved. In the contact-independent mechanism, diverse cytolytic molecules released from the amoeba such as phospholipases, sphingomyelinase, elastase, neuroaminidase, and proteolytic enzymes may induce disruption and apoptosis of host cells in infected loci [3,9,10]. Studies investigating the biological properties and pathogenesis of *N. fowleri* have yet to fully elucidate the pathophysiology of the amoeba, which is necessary to widen our knowledge of the disease mechanisms and the nature of the amoeba.

In this study, we identified a novel cysteine protease inhibitor of *N. fowleri* (NfCPI) by data mining of the *N. fowleri* genomic resource and characterized its biochemical properties. NfCPI shows a dimeric structure and shares biochemical properties with proteins belonging to the cystatin family. Interestingly, the expression of NfCPI is increased during the encystation process and reaches a maximum in the mature cysts of *N. fowleri*, suggesting its essential role in regulation of encystation and formation of mature amoebic cysts.

## 2. Materials and Methods

### 2.1. Cultivation of Naegleria fowleri

*Naegleria fowleri* trophozoites (Carter NF69 strain, ATCC 30215) were cultured in Nelson medium supplemented with 2% heat-inactivated fetal bovine serum (FBS; Gibco, Rockville, MA, USA) and 1% penicillin-streptomycin (Gibco) at 37 °C [11]. The amoeba were maintained by subculturing every three days and used in this study.

### 2.2. Isolation and Cloning of a Gene Encoding NfCPI

The gene encoding NfCPI (Gene ID: NF0117700) was identified by data mining the *N. fowleri* genomic resource (AmoebaDB, http://amoebadb.org/amoeba/, accessed on 16 December 2016). Total RNA of *N. fowleri* was purified using TRIzol reagent (Invitrogen, Carlsbad, CA, USA) and the purified total RNA was treated with RNase-free DNase (Gibco) to digest any contaminating DNA. The cDNA was synthesized from the RNA using the RNA to cDNA EcoDry Premix (Clontech, Mountain View, CA, USA) following the manufacturer’s protocols. The *nfcpi* was amplified using specific primers: 5′-ATGATCAACAACCTTTCCGCACCAGCA-3′ and 5′-TCAAGCAGGTGTATTACAATC- CACTCG-3′. Amplification was performed with a thermal parameter of 95 °C for 10 min; 25 cycles at 95 °C for 1 min, 52 °C for 1 min, and 72 °C for 1 min; and a final extension at 72 °C for 10 min. The PCR product was purified, ligated into T&A cloning vector (Real Biotech Corporation, Banqiao, Taiwan) and transformed into *Escherichia coli* DH5α competent cells. The positive clones were selected by colony PCR, and the nucleotide sequences of the insert were confirmed by automatic DNA sequencing. Analysis of the primary structure of deduced amino acid sequences was performed with the DNASTAR package (DNASTAR, Madison, WI, USA). The signal peptide sequence and potential *n*-glycosylation sites were identified by SignalP5.0 (http://www.cbs.dtu.dk/services/SignalP/, accessed on 21 January 2020) and netNGlyc 1.0 (http:// www.cbs.dtu.dk/services/NetNGlyc/, accessed on 21 January 2020), respectively. The phylogenetic tree was constructed with MEGA6 (http://www.megasoftware.net, accessed on 2 July 2020) using the Maximum Likelihood Estimation (MLE) with Jones–Taylor–Thornton model. The robustness of the nodes was assessed with 1000 bootstrap replications.

### 2.3. Expression and Purification of Recombinant NfCPI

A partial gene of *nfcpi* lacking the signal peptide sequence was amplified with the following primers: 5′-GGATCCCTTCCCATTGCAACCAATGTC-3′, containing a 5′ *Bam*HI site, and 5′-AAGCTTTCAAGCAGGTGTATTACAATC-3′, containing a 5′ *Hind*III site. The amplified product was purified from the gel, and cloned into a T&A vector (Real Biotech Corporation) as described above. The resulting plasmid DNA was digested with *Bam*HI and *Hind*III, cloned into the corresponding restriction sites of pQE-9 expression vector (Qiagen, Valencia, CA, USA), and transformed into *E. coli* M15 (pREP14) competent cells (Qiagen). The selected *E. coli* clone was cultured in Luria Bertani broth, and expression of the NfCPI was induced with 1 mM isopropyl-1-thio-β-_D_-galactopyranoside (IPTG) at 37 °C for 4 h. The cells were collected by centrifugation, suspended in native lysis buffer (50 mM NaH_2_PO_4_, 300 mM NaCl, 10 mM imidazole, pH 8.0), sonicated on ice for 15 min, and centrifuged at 4 °C for 30 min at 12,000 rpm. The supernatant was collected and the NfCPI was purified using the nickel–nitrilotriacetic acid (Ni–NTA) column (Qiagen) following the manufacturer’s instructions. The purification and purity of the NfCPI were analyzed via 15% sodium dodecyl sulfate-polyacrylamide gel electrophoresis (SDS-PAGE) at 150 V for 1 h.

### 2.4. Inhibitory Activity Assay

Inhibitory activity of NfCPI against cysteine proteases was determined by measuring the residual enzyme activity after incubation of each enzyme with NfCPI [12,13]. Cysteine proteases used in this study were papain (Sigma, St. Louis, MO, USA) and two recombinant cathepsin-B and cathepsin-B-like cysteine proteases of *N. fowleri* (NfCB and NfCBL). Recombinant NfCB and NfCBL were produced as described previously [9]. Briefly, 10 nM of each enzyme was incubated with different concentrations of NfCPI (0–100 nM) in 50 mM of sodium acetate (pH 6.0) for 20 min at room temperature (RT). Substrate solution was then added to the mixture, and the residual enzyme activity was determined by measuring the release of fluorescence (excitation at 355 nm and emission at 460 nm) for 30 min at RT with a Fluoroskan Ascent FL (Thermo Fisher Scientific, Vantaa, Finland). Benzyloxycarbonyl-l-leucyl-l-arginine 4-methyl-coumaryl-7-amide (Z-LR-MCA; Peptide International, Louisville, KY, USA) was used as the substrate. Substrate solution was composed of 50 mM sodium acetate (pH 5.0), 10 nM Z-LR-MCA, and 10 mM dithiothreitol (DTT). The concentration of each cysteine protease was determined via active site titration with trans-epoxy-succinyl-l-leucylamido (4-guanidino)butane (E-64; Sigma). The concentration of NfCPI was measured via titration with papain as described previously [14].

### 2.5. Biochemical Properties of NfCPI

Effect of pH on the inhibitory activity of NfCPI was determined by incubating NfCPI (20 nM) with the same concentration of papain, NfCB, or NfCBL in different pH buffers [50 mM sodium acetate (pH 4.0–5.5), 50 mM sodium phosphate (pH 6.0–6.5), or 50 mM Tris-HCl (pH 7.0–8.0)] for 20 min at RT followed by an analysis of the residual activity of each enzyme using the same assay method described above. To determine pH stability of NfCPI, the protein was incubated in different pH buffers at 37 °C for 3 h. The inhibitory activity of NfCPI preincubated under different pH conditions was assayed for papain, NfCB, or NfCBL. The thermal stability of NfCPI was determined by incubating NfCPI at different temperatures (20, 37, and 95 °C) for 0–3 h in 50 mM sodium acetate (pH 5.0). The samples were cooled on ice, and the residual inhibitory activities of sample aliquots against each cysteine protease were determined as described above. In all assays, E-64 was used as a control inhibitor. All the assays were conducted in triplicate, and the mean and standard deviation (SD) were calculated. The equilibrium dissociation constant (*K_i_*) values were calculated using the equation *K**_i_* = IC_50_(1/(1 + [S])/*K*_m_) as described previously [15]. The *K*_m_ of each cysteine protease for the substrate was determined under the same assay conditions in the absence of NfCPI. The *K_i_* values were analyzed via non-linear regression using GraphPad Prism 8 Software (San Diego, CA, USA).

### 2.6. Structural Analysis of NfCPI

In order to analyze the structural properties of NfCPI, electrophoretic analysis of the NfCPI was performed in the presence and absence of SDS or β-mercaptoethanol [β-ME, 5% (vol/vol)] with or without prior heating at 100 °C for 5 min [12,13]. The native molecular size of NfCPI was also analyzed by gel filtration chromatography using a Superdex 200 HR 10/30 column with an Äcta FPLC system (GE Biosciences, Pittsburgh, PA, USA). The purified NfCPI (1 mg) was loaded onto the column, and the collected fractions (0.5 mL) were analyzed by SDS-PAGE, followed by measurement of their inhibitory activities against NfCB and NfCBL. The column was calibrated with gel filtration size marker proteins (Sigma): blue dextran (2000 kDa), β-amylase (200 kDa), alcohol dehydrogenase (150 kDa), albumin (66 kDa), carbonic anhydrase (29 kDa), and cytochrome c (12.4 kDa). The *K_av_* value of each size marker protein was calculated using the equation *K_av_* = (*V_e_* − *V_0_*)/(*V_t_* − *V_0_*), where *V_e_* is the elution volume of protein_,_
*V_0_* refers to the elution volume of blue dextran, and *V_t_* is the total bed volume.

### 2.7. Semiquantitative RT-PCR

To determine the expression profiles of *nfcpi* at different developmental stages of *N. fowleri*, semiquantitative RT-PCR was performed. Encystation of the amoeba was induced by incubating the trophozoites in encystation buffer (120 mM NaCl, 0.03 mM MgCl_4_, 1 mM Na_2_HPO_4_, 1 mM KH_2_PO_4_, 0.03 mM CaCl_2_, 0.02 mM FeCl_2_, pH 6.8) [16] for 60 h at 37 °C. The morphological changes of the amoeba were analyzed with EVOS^®^ XL Core microscope (Life technologies, Carlsbad, CA, USA) and the cells were harvested every 12 h. Total RNA was isolated from *N. fowleri* harvested at each time point using TRIzol (Invitrogen). The cDNA was synthesized using RNA to cDNA EcoDry Premix (Clontech). The transcriptional pattern of *nfcpi* at each developmental stage was analyzed via semiquantitative RT-PCR. The expression profiles of *nfcb*, *nfcbl*, cathepsin Z (*nfcz*) and calpain-like (*nfcalpain-like*) were also analyzed with the following primer sets: *nfcb* (5′-GAGC- TCATGATGATGAGATGCGTGAAT-3′ and 5′-AAGCTTTTAACAACATGCTCCATCA- ATGAAAT-3′), *nfcbl* (5′-GTGAGTAGGAGGTAGCCTTCCA-3′ and 5′-CCTCTTGTTCA- TCACCTTGTTG-3′), *nfcz* (5′-ATGAAAAATCTCTGGCTTGTCTTGCTG-3′ and 5′-TTAA-ACAATTGGAACAGCAAAGGCACA-3′), and *nfcalpain-like* (5′-ATGAACATTCCAGACAAAATTCCG-3′ and 5′-TTACACAACTTTTCTAACTTC-3′). Glyceraldehyde 3-phosphate dehydrogenase of *N. fowleri* (*nfgapdh*) was included as an internal control for relative quantitation of other genes [13].

### 2.8. Production of Monoclonal Antibody

Monoclonal antibody (MAb) specific to NfCPI was produced according to the protocols described previously [17]. The purified NfCPI (50 μg/100 μL) was mixed with an equal volume of Freund’s complete adjuvant (Sigma), and the mixture was injected intraperitoneally (IP) into two six-week-old female BALB/c mice. The mice were boosted twice biweekly via IP injection of the NfCPI (25 μg/100 μL) mixed with an equal volume of Freund’s incomplete adjuvant (Sigma). After the third injection, the NfCPI (5 μg/100 μL) was injected into mice tail intravenously (IV) without any adjuvant. The production of anti-NfCPI antibodies in immunized mice was confirmed by enzyme-linked immunosorbent assay (ELISA) with the NfCPI antigen (1 μg/mL). Goat anti-mouse immunoglobulin (Ig) G (Sigma, 1:5000 dilution) conjugated with alkaline phosphate was used as the secondary antibody and reactions were developed with a substrate buffer containing a p-nitrophenylphosphate (Sigma). Myeloma cells were fused with spleen cells from the immunized mice at a ratio of 1:5 by slowly adding 50% polyethyleneglycol (Sigma). The fused cells were selected with hypoxanthine, aminopterin, and thymidine (HAT) and HT media for two weeks. The supernatant of the hybridoma cells was screened by ELISA as described above. The selected hybridoma cells were cloned by limiting dilution, and the finally selected hybridoma colonies were transferred into 75 cm^2^ tissue culture flasks for mass culture. The supernatant was collected, concentrated, and used for further studies. All the protocols used in the mouse experiments were reviewed and approved by the Animal Research Ethics Committee of the Ajou University School of Medicine (AJ-IBC-17-09-10).

### 2.9. Immunoblot

Expression profile of NfCPI in the *N. fowleri* trophozoite and cyst was analyzed by immunoblotting. The lysates of *N. fowleri* trophozoites and cysts were prepared by repeated freezing and thawing procedures in RIPA Lysis and Extraction buffer (Thermo Fisher Scientific) followed by sonication on ice. The lysates of *N. fowleri* trophozoites (20 μg) and cysts (20 μg) were separated by 15% SDS-PAGE and transferred electrophoretically onto the nitrocellulose membrane. The membrane was blocked with PBS supplemented with 0.05% Tween 20 (PBST) and 5% skim milk for 1 h and incubated with anti-NfCPI MAb (1:1000 dilution in 5% skim milk) or IgG purified from non-immunized mouse serum at RT for 2 h. After several washes with PBST, the membrane was incubated with horseradish peroxidase (HRP)-conjugated anti-mouse IgG (Sigma, 1:1000 dilution in 5% skim milk) at RT for 2 h. The membrane was washed with PBST several times and immunoreactive bands were visualized with 4-chloro-1-naphthol (Sigma).

### 2.10. Immunofluorescence Assay (IFA)

Localization of NfCPI at different developmental stages of *N. fowleri* was determined by IFA with a slightly modified method [18]. *N. fowleri* trophozoites and cysts on the coverslip were fixed with 10% formaldehyde for 10 min at room temperature, permeabilized in 1% ammonium hydroxide, washed in Tween 20 for 5 min, and then washed extensively with 0.82% physiological saline. Anti-NfCPI MAb (1:100 diluted in PBS) was added to the trophozoites and cysts and incubated overnight at 4 °C. After washing with PBS several times, fluorescent isothiocyanate (FITC)-conjugated goat anti-mouse IgG (Sigma, 1:200 dilution) was added and incubated at 4 °C for 2 h. Fluorescence was visualized in the optical section produced by confocal laser scanning microscopy (Carl Zeiss, Oberkochen, Germany).

### 2.11. Statistical Analysis

All experiments were performed in triplicate. Statistical significance was evaluated by one-way ANOVA, followed by one-tailed Student’s *t* test. Differences in mean values at *p* < 0.05 were considered statistically significant.

## 3. Results

### 3.1. Sequence Analysis of NfCPI

The open reading frame of *nfcpi* consisted of 411 bp encoding 136 amino acid residues with a predicted molecular mass of 14.7 kDa. Analysis of deduced amino acid sequence revealed that NfCPI had a typical N-terminal signal peptide sequence of 29 amino acids (MINNLSAPAAATTLIFILALLITFSNILA) and two potential *N*-glycosylation sites at 4NLS6 and 46NQT48. Multiple sequence alignment of amino acid sequences of NfCPI and related proteins from other organisms revealed low sequence identity. However, the Gln-Val-Val-Ala-Gly (QVVAG) motif, a tightly conserved motif of cystatin family, was identified in NfCPI (Figure 1A). Two C-terminal cysteine residues that form a disulfide bond in the cystatin C, D, and S superfamily [19,20] were also detected in the NfCPI. Phylogenetic analysis of NfCPI with other related proteins also revealed that NfCPI formed a large cluster with cystatin C, D, and S superfamily proteins (Figure 1B).

### 3.2. Expression and Purification of NfCPI

A fragment of *nfcpi* lacking the 29 N-terminal amino acid sequences that correspond to signal peptide sequence was expressed in *E. coli*. The NfCPI was expressed as a soluble protein with an approximate molecular mass of 12 kDa, which coincided with the expected molecular mass of the protein calculated with the deduced amino acid sequences. The recombinant protein was purified using Ni–NTA affinity chromatography (Figure 2A).

### 3.3. Dimeric Structure of NfCPI

Electrophoretic analysis of NfCPI under non-reducing conditions revealed that the protein formed a dimeric structure. The size of NfCPI in non-reducing PAGE was approximately 24 kDa (Figure 2B). The size was not affected by SDS. Meanwhile, β-ME affected the electrophoretic movement of NfCPI to a size of about 12 kDa in the presence or absence of heating, suggesting the role of inter-disulfide bonds in the dimeric structure formation of NfCPI. To further analyze the native size of NfCPI, gel filtration chromatography was performed. The NfCPI was mainly detected in the fractions 26 to 28, which corresponded to the predicted molecular size of 24 kDa (Figure 2C).

### 3.4. Inhibitory Activity of NfCPI

Inhibitory activity of NfCPI against several cysteine proteases including papain, NfCB, and NfCBL was analyzed. NfCPI inhibited the enzyme activities of all the tested enzymes in a dose-dependent manner. However, it showed greater inhibition against NfCB and NfCBL, endogenous cysteine proteases of *N. fowleri*, than papain (Figure 3A). The *K_i_* values also suggested that NfCPI inhibited NfCB and NfCBL more effectively than papain in the picomolar range (Table 1). NfCPI inhibited the tested enzymes under a broad range of pH values, suggesting that the interactions between NfCPI and the enzymes were not influenced by pH (Figure 3B). NfCPI was stable under different pH conditions resulting in no significant change in its inhibitory capacity (Figure 3C). The thermal stability analysis revealed the heat-labile features of NfCPI. The protein was stable at 20 and 37 °C at least for 3 h, but it rapidly lost its inhibitory activity when incubated at 95 °C (Figure 3D).

### 3.5. Expression Pattern and Cellular Localization of NfCPI

The trophozoites of *N. fowleri* were successfully transformed to cysts following incubation at 60 h in the encystation medium (Figure 4A). RT-PCR was performed using the RNAs isolated at each stage during the encystation process to investigate the transcription profiles of *nfcpi* from trophozoites to cysts. The expression of *nfcpi* was not detectable at the trophozoite stage, but it gradually increased starting at 12 h post-incubation, suggesting that *nfcpi* was mainly expressed during encystation and in mature cysts (Figure 4B). Expression profiles of several endogenous cysteine proteases, including *nfcb*, *nfcbl*, *nfcz* and *nfcalpain-like,* were also analyzed. For *nfcb*, *nfcbl*, and *nfcz*, the highest levels of expression were identified in trophozoites and the gene expression level declined eventually as the trophozoites transformed into cysts. Meanwhile, *nfcalpain-like* was not expressed in trophozoites. However, its expression was gradually increased at 12 h post-incubation, which coincided with the expression pattern of *nfcpi*. In order to confirm the expression profile of NfCPI at different developmental stages, an immunoblot using lysates of *N. fowleri* trophozoites and cysts and anti-NfCPI MAb was performed. Immunoblot results revealed elevated levels of NfCPI in cysts but not in trophozoites (Figure 4C). IFA also revealed that NfCPI was mainly identified in the cytosol of cysts in clusters that were not further identified (Figure 4D).

## 4. Discussion

Proteases of protozoan parasites play essential biological functions in the pathophysiology and pathobiology of organisms [21]. Several classes of proteases of *N. fowleri* have attracted great attention because of their key roles in the physiology and pathogenesis of amoeba [9,22,23,24,25,26]. Although these enzymes play pivotal roles in physiology and pathobiology of amoeba, a stringent regulation of their activities is also essential to control or modulate normal biological processes in amoeba. Studies have reported the essential role of parasitic protozoan cysteine protease inhibitors in the survival and infectivity of parasites in hosts by regulating endogenous or exogenous proteases [27,28,29,30]. Previously, we identified and characterized a novel stefin-family cysteine protease inhibitor of *N. fowleri*, known as fowlerstefin [13]. In this study, we characterized the second cysteine protease inhibitor of *N. fowleri*, NfCPI.

NfCPI is a typical cysteine protease inhibitor belonging to the superfamily of cystatin C, D, and S superfamily proteins. The QVVAG motif, which forms a hairpin loop involved in direct interaction with target enzymes [19,31],was conserved in NfCPI. Two C-terminal cysteine residues that may form disulfide bonds were also found in NfCPI. Non-reducing PAGE and gel infiltration analyses suggested a dimeric structure of NfCPI, which further support the involvement of the two cysteine residues in inter-disulfide bridge formation. Dimeric structures linked by inter-disulfide bonds are usually identified in extracellular cystatins [19,20]. Although NfCPI has a potential signal peptide sequence at its N-terminal region, it is not clear whether it is actively secreted outside of *N. fowleri* because the protein was mainly detected in the cytoplasmic region of the cysts.

NfCPI effectively inhibited cysteine proteases such as NfCB, NfCBL, and papain under a broad range of pH conditions, suggesting that NfCPI is a functional protein. Although NfCPI effectively inhibited two endogenous cysteine proteases of *N. fowleri*, NfCB and NfCBL, it is likely that the two enzymes may not represent the primary targets of NfCPI since the transcription patterns of the two counterpart proteins did not coincide with that of *nfcpi*. The transcription levels of *nfcb* and *nfcbl* were the highest in trophozoites and decreased significantly during the encystations of the amoeba. In contrast, the transcriptional level of *nfcpi* was gradually increased in the amoeba during encystation and reached a maximum in the mature cysts. Immunoblot and IFA results also supported its high level of expression in cysts. Data mining of the *N. fowleri* genomic resource (AmoebaDB) shed light on diverse cysteine proteases of the amoeba. We analyzed stage-specific transcriptional patterns of the genes encoding candidate cysteine proteases of *N. fowleri* and found that transcription profile of Nfcalpain-like enzyme (*nfcalpain-like*) was identical to that of *nfcpi*. The expression patterns of the other candidate enzymes including *nfcb*, *nfcbl*, other homologs of *nfcb*, and *nfcz* were not consistent with the expression profile of *nfcpi*. These results suggested that Nfcalpain-like enzyme could be a potential primary target for NfCPI. Calpains are Ca^2+^-dependent cysteine proteases belonging to clan CA protease. They play essential roles in a range of crucial cellular processes in diverse organisms including humans, animals, plants, nematodes, and protozoa [32,33]. Although the calpain family of enzymes is well characterized in animals and plants, their role in unicellular eukaryotes including *N. fowleri* is unknown. We tried the expression of recombinant Nfcalpain-like enzyme to determine the functional relevance of NfCPI as a calpain inhibitor. However, unfortunately, we failed to obtain an enzymatically active recombinant Nfcalpain-like protein. Further studies are necessary to determine whether Nfcalpain-like is a genuine target of NfCPI.

The biological function of NfCPI is not clear. Given its increased expression in amoeba under encystation and that it reaches maximum levels in mature cysts, NfCPI may play an important role in encystation and cyst formation of *N. fowleri*. Encystation is an essential step in differentiation of amoeba during metabolic and transcriptional changes, which initiate or induce large modifications of subcellular organelles as well as morphological changes of the amoeba itself [34,35,36]. The functional significance of proteases in amoeba cyst formation has been partially determined in few previous studies. *Entamoeba invadens* EiCP-B is specifically expressed near the cyst wall during encystation [37]. A serine protease M17 leucine aminopeptidase of *Acanthamoeba castellanii* (AcLAP) plays an important role in mature cyst formation [38]. In *Naegleria* spp., the molecular mechanisms associated with encystation have not been clearly understood. The role of proteases in encystation of the amoeba is also unknown. Only a few molecules, including ubiquitin-like Atg8 and enolase, have been reported to be associated with encystation [39,40]. Given the specific expression of NfCPI during encystation of *N. fowleri*, it could play a critical role in regulating the proteolytic activities of cysteine proteases, mediating the encystation process or mature cyst formation in amoeba.

## 5. Conclusions

NfCPI is a functional cystatin-family protease inhibitor of *N. fowleri*. Specific expression of the protein during encystation and at mature cyst of the amoeba suggests its potential roles in encystation and cyst formation of *N. fowleri*. Given the highly limited knowledge of the molecular mechanism underlying the induction or modulation of encystation of *N. fowleri*, further studies exploring the role of proteases and protease inhibitors are necessary to obtain an in-depth insight into the molecular mechanism of encystation of the amoeba. Studies investigating the molecular mechanism of excystation of the amoeba will also contribute to our understanding of the biological properties and pathogenicity of the amoeba.

## Figures and Tables

**Figure 1 pathogens-10-00388-f001:**
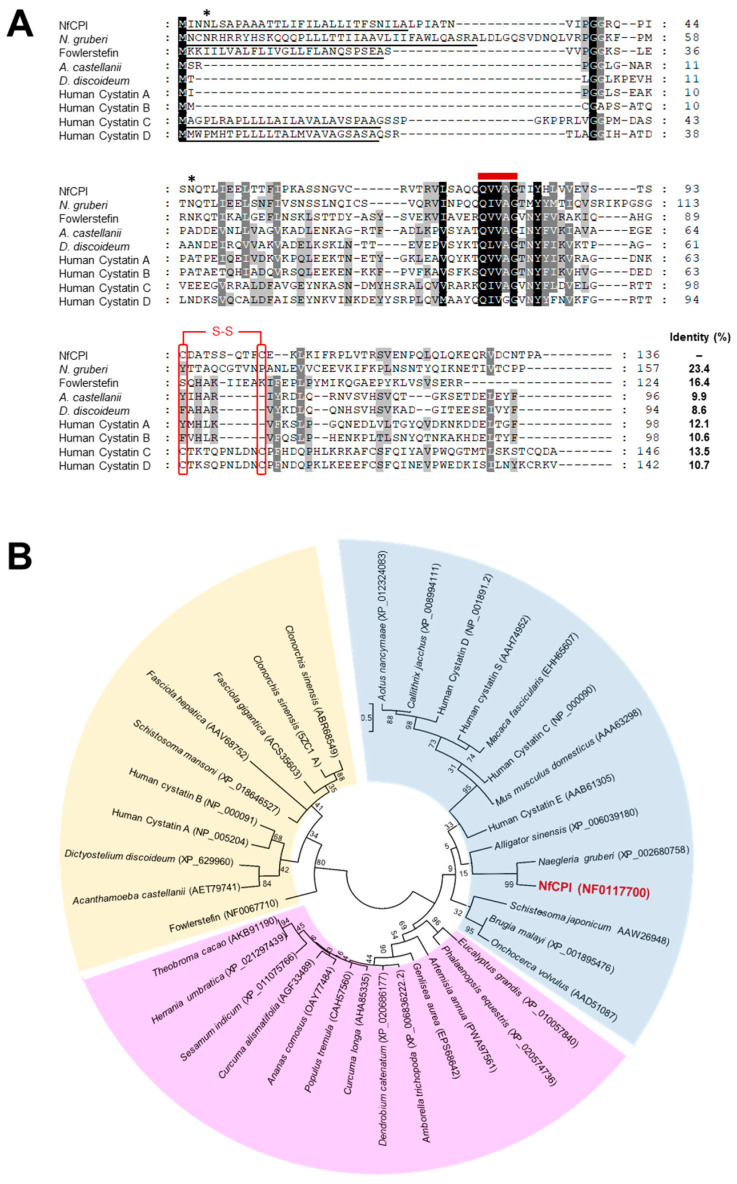
Multiple sequence alignment and phylogenetic analysis. (**A**) Multiple sequence alignment. The deduced amino acid sequence of *N. fowleri* (NfCPI) was aligned with the sequences of related proteins derived from other protozoa and humans. The Gln-Val-Val-Ala-Gly (QVVAG) cystatin motif is presented as a bold red line on the sequences. The predicted N-terminal signal peptide sequences are underlined. The predicted putative *N*-glycosylation sites in NfCPI are marked by asterisks. A potential disulfide bridge is presented in brackets and the associated cysteine residues are boxed in red color. The shading displays the degree of identity among the sequences. Sequence identity between NfCPI and other related proteins is shown on the right side. Sequence identity between NfCPI and other related proteins is shown on the right side. *Naegleria gruberi* CPI (XP_002680758.1), Fowlerstefin (NF0067710), *Acanthamoeba castellanii* CPI (AET79741), *Dictyostelium discoideum* CPI (XP_629960), Human cystatin A (NP_005204), Human cystatin B (NM_000100), Human cystatin C (NP_000090), and Human cystatin D (NP_001891) are included in the alignment. (**B**) The phylogenetic tree was constructed by the maximum likelihood method using the MEGA6 program. Yellow, cystatin A and B clades; blue, cystatin C, D, and S clades; pink, CPIs from plants. Numbers on the branches show bootstrap proportion (1000 replicates).

**Figure 2 pathogens-10-00388-f002:**
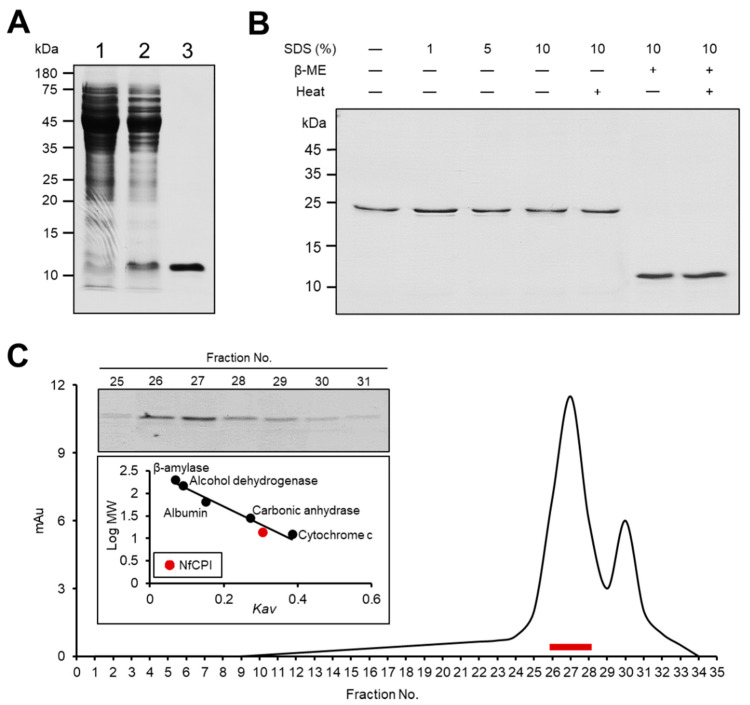
Expression of recombinant NfCPI and structural analysis. (**A**) The NfCPI was purified using Ni–NTA affinity chromatography and analyzed via sodium dodecyl sulfate-polyacrylamide gel electrophoresis (SDS-PAGE). Lane 1, non-induced *E. coli* lysate; lane 2, isopropyl-1-thio-β-d-galactopyranoside (IPTG)-induced *E. coli* lysate; lane 3, Ni–NTA affinity purified NfCPI. (**B**) The NfCPI was mixed with different concentrations of SDS (0–10%, *v*/*v*) and β-ME with or without heating and analyzed via SDS-PAGE. (**C**) Gel filtration chromatography analysis. The NfCPI was loaded to Superdex 200 HR 10/30 column and fractions (0.5 mL) were collected. The collected fractions were analyzed by SDS-PAGE and their inhibitory activities against NfCB and NfCBL were assayed. The fractions expressing strong inhibitory activity (fractions 26 to 28) are represented as red bars and confirmed by SDS-PAGE. The *K_av_* value of NfCPI (red dot) was calculated by comparing with those of size markers (black dots).

**Figure 3 pathogens-10-00388-f003:**
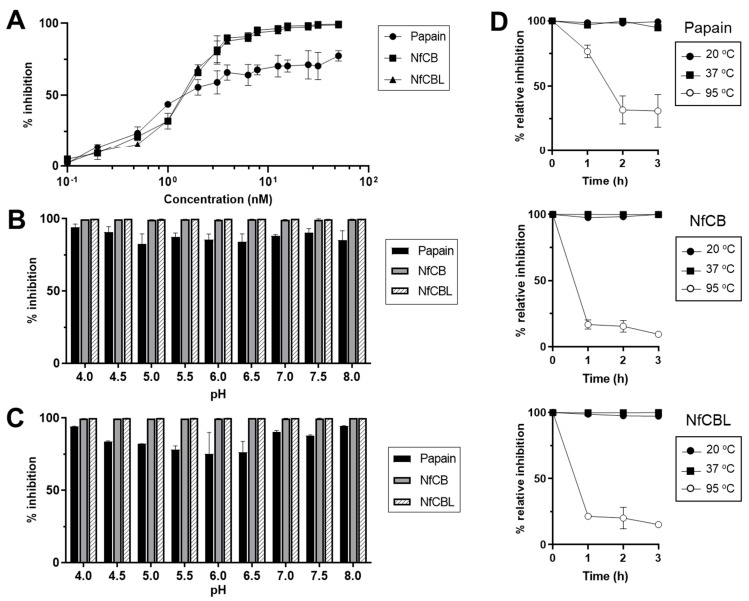
Inhibitory activity of NfCPI. (**A**) Inhibitory activity of NfCPI against cysteine proteases. Different concentrations of NfCPI (0–100 nM) were incubated with each cysteine protease (10 nM) and the residual enzyme activity of each enzyme was assayed. (**B**) pH dependency. NfCPI was incubated with papain, NfCB and NfCBL in different pH buffers and the residual enzyme activity was analyzed. (**C**) pH stability. NfCPI was incubated in different pH buffers for 3 h and its inhibitory activity against cysteine proteases was assayed. (**D**) Thermal stability. NfCPI was incubated at 20, 37, and 95 °C for the indicated time points, respectively. The inhibitory activity of NfCPI against cysteine proteases was analyzed. All experiments were performed in triplicate and the mean and standard deviation (SD) values are presented.

**Figure 4 pathogens-10-00388-f004:**
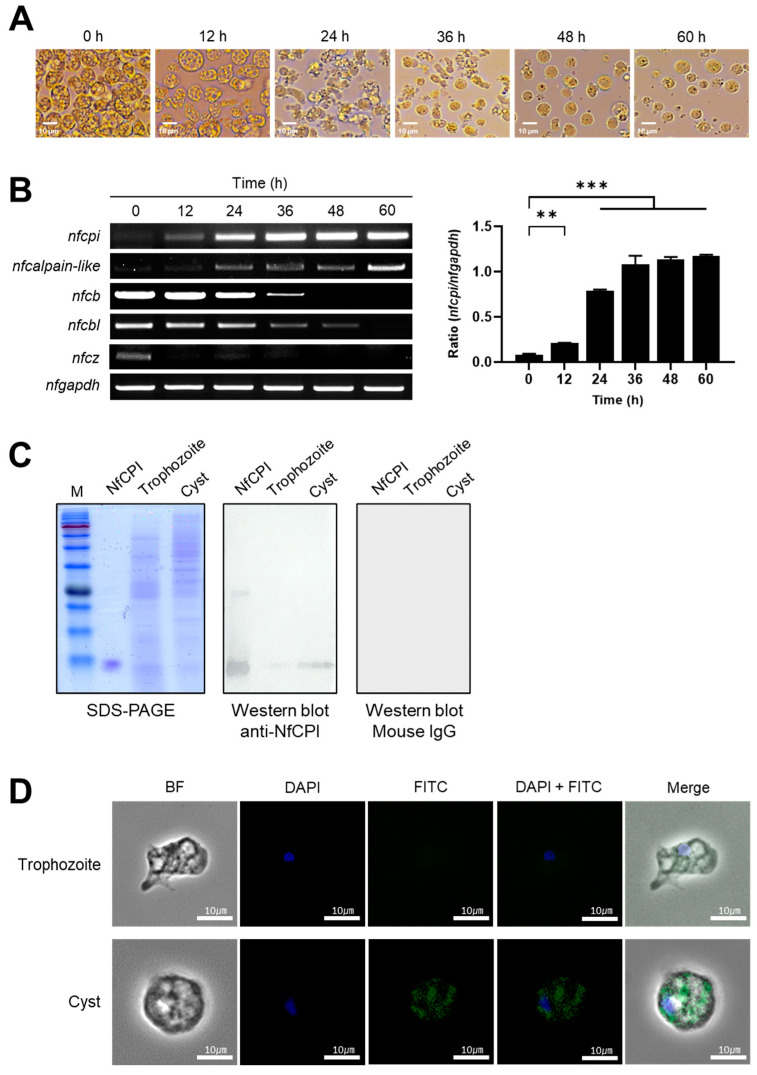
Expression pattern of NfCPI in *N. fowleri*. (**A**) The morphological changes of *N. fowleri* during encystation. *N. fowleri* trophozoites were incubated in encystation medium and morphological changes of the amoeba were analyzed at the indicated time points by microscopy. (**B**) Semiquantitative RT-PCR. The transcription profiles of *nfcpi*, *nfcb*, *nfcbl, nfcz,* and *nfcalpain-like* were analyzed during encystation (left panel). Graphs show the mean ± SD densitometric ratios of *nfcpi* and *nfgapdh* of three independent experiments (right panel). Significance of the data was analyzed by using one-tailed *t* test, ** *p* < 0.005, *** *p* < 0.001. (**C**) Immunoblot analysis. Expression profile of NfCPI in trophozoites and cysts was determined by immunoblot with anti-NfCPI MAb or non-immunized mouse IgG. (**D**) Localization. Immunofluorescence Assay (IFA) was performed with anti-NfCPI MAb. *N. fowleri* trophozoites and cysts were fixed on cover slip and probed with anti-NfCPI MAb and FITC-conjugated anti-mouse IgG. The slide was observed with a confocal laser scanning microscope.

**Table 1 pathogens-10-00388-t001:** *K_i_* values of NfCPI against cysteine proteases.

Enzyme	*K*_*i*_ (nM) ± SD
Papain	0.259 ± 0.034
NfCB	0.193 ± 0.032
NfCBL	0.144 ± 0.003

## Data Availability

The original data in the present study are available from the corresponding author upon request.

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
