# Peer review of "A Novel Cysteine Protease Inhibitor of Naegleria fowleri That Is Specifically Expressed during Encystation and at Mature Cysts"

_pathogens, 2021, doi:10.3390/pathogens10040388_

Round 1

Reviewer 1 Report

This article wrote by Huang Giang le et al. is really well-constructed and main objectives are clear and experiments are adapted to following this goal. Experiments are multiple and this article is suitable for publication with minor revision.

The end of the introduction need to be rephrase I not understand before the paragraph lines 60-65. What did you do to retrieve this novel cysteine? More rational could add in the introduction (1/2 sentences)

Author need to do some minor revisions or typo mistakes:

In line 41: Seizure => Could you precise?

In line 46: bold for cases

-line 78: Moutain view in bold and nfci not format as others

Precision for the PCR you only used 25 cycles?

In line 91: authors used MEGA6 could precise parameter used for the alignment and for the maximum likelihood tree.

In line 108 authors need to add parameter used for the electrophoresis.

In line 114 please spell NfCB and NfCBL

In lines 175-176 authors said G3PDH was included as an internal control. Authors could precise ? Is it for calculate the relative expression or “just” to verify the presence of enough DNA/RNA materials.?

In line 210 please spell PBST

In the figure 3D it could be interesting to know the activity at 20°C.

This test is mandatory to understand the pathogenesis of amoebas specially to understand amoebas and their environmental impact.

Line 301 medium or media?

In line 327 please replace protozoan by protist.

Please rephrase the sentence line 330

Line 368 protist

Reviewer 2 Report

The manuscript is very interesting and well-written. The topic of Naegleria infection is important and as the authors rightly state, especially so in light of global warming. The research contributes towards our understanding of Naegleria encystation and offers a path to potential drug targets.

The methods are appropriate and described very well and sufficient detail. The language throughout is of the highest standard and the composition is easy to understand. All concepts are adequately explained and backed up with the correct citations.

All expected controls have been performed and the results are clearly presented.

I only have minor comments to improve the manuscript:

P2 L42: meningoencephalitis

P5 L206, 207, 210: “skimmed milk”

P6 Figure 1B: I would indicate NfCPI in bold or draw a box around it to make it stand out in the tree.

P9 L303: I would reword “…was not critical…” to “…was not detectable…” because the author didn’t perform functional studies

P9 L315: “…in the cytosol of cysts with clustered forms” – I would say “…in the cytosol of cysts in not further identified clusters.”

P10 Figure 4A: The Figure would be perfect with a scale bar added to the microscopic images.

P10 Figure 4C: The Western blot looks a bit grainy and might have been digitally enhanced. Whereas I have no doubt that this reflects the true experiment and maybe due to weak signal, an indication of exposure time or if the image was digitally enhanced would help.

P11 L322: “the amoeba”

P11 L347: “…the protein was not detected in the culture medium.” The relevant experiments are not shown or mentioned in the results section. Please insert a sentence describing how you tried to detect the protein in the culture supernatant. I completely understand that the antibody might not have been strong enough to detect and the concentration of protein might have been too low for detection.

Author Response

This manuscript is a resubmission of an earlier submission. The following is a list of the peer review reports and author responses from that submission.